# KuaiSim: A Comprehensive Simulator for Recommender Systems

**Kesen Zhao[1], Shuchang Liu[2*], Qingpeng Cai[2†], Xiangyu Zhao[1†],**
**Ziru Liu[1], Dong Zheng[2], Peng Jiang[2], Kun Gai[3]**
[1]City University of Hong Kong, [2]Kuaishou Technology, [3]Unaffiliated
{kesenzhao2-c,ziruliu2-c}@my.cityu.edu.hk
{liushuchang, caiqingpeng, zhengdong, jiangpeng}@kuaishou.com
xianzhao@cityu.edu.hk, gai.kun@qq.com

## Abstract

Reinforcement Learning (RL)-based recommender systems (RSs) have garnered considerable attention due to their ability to learn optimal recommendation policies and maximize long-term user rewards. However, deploying RL models directly in online environments and generating authentic data through A/B tests can pose challenges and require substantial resources. Simulators offer an alternative approach by providing training and evaluation environments for RS models, reducing reliance on real-world data. Existing simulators have shown promising results but also have limitations such as simplified user feedback, lack of consistency with real-world data, the challenge of simulator evaluation, and difficulties in migration and expansion across RSs. To address these challenges, we propose KuaiSim, a comprehensive user environment that provides user feedback with multi-behavior and cross-session responses. The resulting simulator can support three levels of recommendation problems: the request level list-wise recommendation task, the whole-session level sequential recommendation task, and the cross-session level retention optimization task. For each task, KuaiSim also provides evaluation protocols and baseline recommendation algorithms that further serve as benchmarks for future research. We also restructure existing competitive simulators on the KuaiRand Dataset and compare them against KuaiSim to further assess their performance and behavioral differences. Furthermore, to showcase KuaiSim's flexibility in accommodating different datasets, we demonstrate its versatility and robustness when deploying it on the ML-1m dataset. The implementation code is available online to ease reproducibility [3].

## 1   Introduction

Reinforcement Learning (RL)-based recommender systems (RSs) have drawn considerable attention in both academia and industry [45, 49, 5, 3, 4, 52, 51, 55]. They regard the recommendation procedures as sequential interactions between users and agents, and learn an optimal recommendation policy that maximizes the long-term cumulative reward from users. While it is theoretically superior to standard learning-to-rank methods [2], it is usually sub-optimal to evaluate the RL model using offline data, because of the missing online exploration and the impossibility of counterfactual evaluation. As an alternative, one may directly deploy RL models in online environments and let them interact with real users. In practice, this turns out to be challenging and resource-intensive [58], since an untrained or premature recommendation model can adversely impact the user experience and lead to undesirable real-time data, subsequently affecting the model's training performance [20]. In order to

---

*The first two authors contributed equally to this work
†Corresponding author

[3]https://github.com/Applied-Machine-Learning-Lab/KuaiSim

37th Conference on Neural Information Processing Systems (NeurIPS 2023) Track on Datasets and Bechmarks.

mitigate these challenges, the research of user simulators has emerged recently to serve as a pre-online verification method for RL models [8, 14, 11, 37, 44, 40, 57]. Intuitively, simulators offer a practical solution for simulating user responses, allowing for the training and evaluation of RS models that are challenging to assess using offline data alone. And it enables researchers to iteratively improve RS models without relying exclusively on real-time user interactions.

Despite the effectiveness of existing simulators in some specially designed recommendation scenarios [14, 44], they are still far from the real-world environment in several aspects, which limits their effectiveness in evaluating online learning models: 1) While the most standard setup simulates single immediate feedback, user responses in most web services are multi-behavior in essence, for example, in short-video recommendation platforms, users can click, like, forward, or download. 2) In addition to the aforementioned immediate responses, a user may leave the app and then come back, generating a leave signal and a retention signal, but existing simulators do not address these long-term or delayed behaviors. The retention signal, in particular, is closely related to some crucial evaluation metrics (i.e., the daily average users) of online systems [54, 27], which could be identified as an essential building block that is still seriously underestimated in the research field. 3) Many rule-based simulators [14] are specially constructed for certain recommendation scenarios that may restrict their accommodation towards other recommendation services. 4) More recent simulators [44, 41] addressed the previous issue through pretraining of simulators according to log data, but during online interactions, they need to sample a user according to a pretrained user generator in addition to the user response model. This may further amplify the inconsistency between the simulated environment and the real-world data distribution. 5) Finally, simulators are capable of evaluating recommendation models, but there is still limited research on how to evaluate a simulator. In other words, we can only be certain of the superiority of an RL method when it achieves better results on the simulator and the simulator is consistent with the real-world environment.

To meet this demand, we propose KuaiSim, a comprehensive simulator that provides a user response environment at three distinct task levels: the request level recommendation task addresses the multi-behavior feedback and list-wise evaluation, the whole-session level sequential recommendation task addresses the long-term reward optimization under the standard RL setting, and cross-session level recommendation task address the retention optimization problem. The resulting simulator consists of a user immediate response model that generates feedback for each recommendation, a user leave model that specifies the end of a session, and a user retention model that determines how long the user returns to the system and starts a new session. To ensure its consistency with the real-world environment, KuaiSim uses the log data to pretrain the user response models and engage user sampling during simulation. This also allows for flexible adaptation on other datasets as long as the required data format suffices. In summary, our contribution can be summarized as follows:

- We propose a comprehensive simulator, KuaiSim, that covers three levels of recommendation tasks to encompass a majority of recommendation challenges. We also provide comparisons of various competitive algorithms for each task as benchmarks to boost future research work.

- Our refined simulator construction and evaluation process is user-friendly and flexible, and we showcase the data migration capabilities of KuaiSim on both the original dataset KuaiRand and the widely available ML-1m benchmark dataset.

- Additionally, we conduct a comparative analysis of KuaiSim against existing simulators, including RecoGym [38], RecSim [14], RL4RS [44], and VirtualTaobao [41]. The results demonstrate that our simulator excels in its ability to approximate real-world environments.

## 2 Preliminary

### 2.1 List-wise Recommendation

The list-wise recommendation problem focuses on the interconnected nature of items within a presented list, as highlighted in our previous studies [56, 59, 60]. The main objective is to understand and learn from the distinctions between including or excluding specific items in the exposed list [2, 32]. Recent research has indicated that in multi-stage recommendation systems, re-ranking models can effectively capture item correlations due to the reduced candidate set size, enabling the utilization of powerful neural models [29]. In the past few years, there has been an ongoing discussion regarding the generative perspective of list-wise recommendation [16, 29]. To address the challenge of the vast combinatorial output space for lists, the generative approach directly models the distribution of recommended lists and generates entire lists using deep generative models. For instance, ListCVAE [16] employs Conditional Variational Autoencoders (CVAE) to capture positional biases and item interdependencies within the list distribution. However, our previous studies [29] have revealed that

Table 1: A comparison between KuaiSim and other simulators.

| Simulators | Real dataset | Recommendation task | | |
|---|---|---|---|---|
| | | Request-level | Whole-session | Cross-session |
| RecoGym | | | ✓ | |
| RecSim | | | ✓ | |
| RL4RS | ✓ | ✓ | ✓ | |
| Virtual-Taobao | ✓ | | ✓ | |
| KuaiSim | ✓ | ✓ | ✓ | ✓ |

ListCVAE struggles with accuracy-diversity trade-offs. One limitation of these methods is that they fail to consider the broader impact of a single recommendation on the entire session.

## 2.2 Sequential Recommendation

The session-based recommendation (SBR) considers the beginning and end of an interaction session. Our ultimate objective is to optimize the overall future reward of the user session, which aligns more with the notion of partial SBR as defined in [44]. In our settings, the prevailing solution relies on the Markov Chain assumption to model the dynamic transitions in user-system interactions. The primary challenge in this line of research is how to construct a representative user state based on extensive historical data [22, 25]. Initial solutions to the recommendation problem utilized collaborative filtering techniques [7, 18, 24, 35], while subsequent approaches incorporated deep neural networks [48] such as Recurrent Neural Networks [15], Convolution Networks [43] and Self-Attention [42, 53, 21] to enhance the model's expressive power. These techniques aim to capture the rich and intricate information contained in user and item features, as well as their interaction histories. However, the common underlying principle of these methods is to accurately encode long-term histories without directly optimizing for long-term user rewards.

## 2.3 Retention Optimization

The objective of the next session recommendation is to suggest the next session, based on the analysis of inter-session dependencies within historical sessions [46]. In addition to considering dependencies within individual sessions, the inclusion of inter-session dependencies, such as user retention, has been proven to be crucial for improving the overall performance of recommendations by our previous works [54, 3]. Multi-level session data exhibits a hierarchical structure, encompassing multiple levels, including interaction and session levels. Within this framework, both the dependencies within each level and those across different levels play a pivotal role in influencing subsequent recommendations. For instance, the recommended results from the previous session can impact the time interval before the user initiates the next session (return time). Consequently, effectively capturing and accurately modeling inter-level dependencies, while optimizing user retention, poses a significant challenge [3]. Unfortunately, there is a scarcity of research addressing this area. In response to this need, our simulator offers a unique capability to predict the return time of users, providing a valuable tool for conducting in-depth research in this under-explored domain.

## 2.4 Existing Simulators

The efficacy of reinforcement learning in recommendation systems is attributable to its ability to learn from user feedback and adapt to evolving user preferences over extended periods [1, 12, 30, 61]. However, the application of reinforcement learning to real-world tasks can be challenging due to the large volume of data involved, which can result in high sampling costs [41]. To address this challenge, researchers have developed simulators for recommender systems using reinforcement learning (RL) algorithms. In this paper, we provide an overview of the existing four simulators.

**RecoGym** [38] RecoGym overcomes the issue of exploding variation and allows for the simulation of user reactions to arbitrary recommendation policies. It is based on a cycle of collecting performance data, developing a new version of the recommendation model, A/B testing, and rolling out, which shares similarities with the RL setup.

**RecSim** [14] Recsim is a configurable simulation platform, which supports sequential interaction with users and allows for easy configuration of environments with varying assumptions about user preferences, item familiarity, user latent state and dynamics, and choice models.

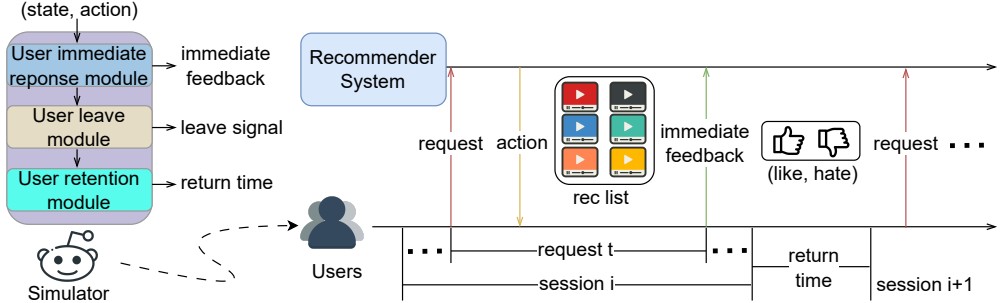

Figure 1: The workflow of simulator and a common MDP setting for user-system interaction.

**RL4RS** [44] Current research on RL-based recommender systems (RL-based RS) lacks a properly validated simulation environment, advanced evaluation methods, and a real dataset, leading to a reality gap. To address this, the RL4RS dataset is introduced, which aims to bridge the reality gap in current RL-based RS research. The dataset is collected from a popular game by NetEase Games and is anonymized using a three-phase anonymization procedure.

**Virtual-Taobao** [41] Virtual-Taobao is a simulator that is trained using historical customer behavior data. To enhance the simulation accuracy, it utilizes GAN-SD (GAN for Simulating Distributions) and MAIL (Multi-agent Adversarial Imitation Learning) for generating customer features with better distribution matching and more generalizable customer actions.

But all these simulators ignore the optimization of long-term feedback, such as user retention, and mostly support only a single recommended task. In Table 1, we summarize the characteristics of existing works and our KuaiSim in terms of datasets and tasks. It can be seen that our KuaiSim benchmark is the only one that meets all requirements.

## 3 Methodology

This section begins by providing a comprehensive explanation of the fundamental definitions and settings pertaining to our three tasks: request level, whole-session level, and cross-session level. Subsequently, we delve into the intricacies of constructing our simulator, offering a detailed account of the process. Lastly, we present an introduction to the dataset we employ and conduct a thorough analysis of the KuaiRand dataset.

### 3.1 Multi-level Reinforcement Learning-based Recommendation

We consider the RL-based recommendation task as a sequential interaction between a recommender system (agent) and users (environment) and emphasize the multi-behavior and cross-session nature of this interaction. Specifically, users may provide short-term feedback in different formats such as clicks, views, and likes; and they may also provide delayed long-term feedback like the retention signal that can only be observed when users return to the system. We provide the overview of this comprehensive interaction paradigm as Figure 1, which is a cross-session multi-behavior user-system interaction flow. Formally, in the $t$-th interaction step in a specific session, the recommendation system receives a user request $\mathcal{O}_t$ and generates a recommendation $\mathcal{A}_t$ as action. Here we assume that each user request consists of a set of static user profile features $\mathcal{U}$ and the most recent interaction history $\mathcal{H}_{:t-1}$ that dynamically changes through time. We also assume a candidate item set $\mathcal{C}$, so a typical recommendation action has $\mathcal{A}_t \in \mathcal{C}^K$, which is a list of size $K$. And a user session typically refers to the continuous interactions of the user from the starting point when opening the app until the exit. On the other side of the flow, the user receives the recommendation and provides the user feedback $\mathcal{Y}_t$ of three types: 1) The immediate feedback $\mathcal{Y}_t^{(I)} \in \mathbb{R}^{b \times K}$ directly reveals the user's preference of the recommended items, and $b$ represents the number of behavior signal types; 2) The leave signal $\mathcal{Y}_t^{(L)} \in \{0, 1\}$ specifies whether the user stops the current session and exits the app; 3) If the user leaves the current session, there is an additional return time (i.e. retention) signal $\mathcal{Y}_t^{(R)}$ that indicates how long the user will come back and start a new session. Then, we can formulate the three-level learning tasks with different learning goals and each focuses on a specific research question, namely, the request level, the session-level, and the cross-session level.

---
**Algorithm 1** Step — the detail workflow of KuaiSim.

---

**Input Format**: observation $\mathcal{U}$, $\mathcal{H}_{:t-1}$, and recommendation action $\mathcal{A}_t$

**Output**: immediate feedback $\mathcal{Y}_t^{(I)}$, leave signal $\mathcal{Y}_t^{(L)}$, and retention signal $\mathcal{Y}_t^{(R)}$ (cross-session)

**The user immediate response module:**

1:     User history encoding $\mathbf{h}_t \leftarrow$ Transformer $(\mathcal{U}, \mathcal{H}_{:t-1})$
2:     Ground truth user state $\mathbf{s}_t \leftarrow \mathbf{h}_t \oplus \mathcal{U}$
3:     Behavior attention $w_t \leftarrow \text{DNN}(\mathbf{s}_t)$
4:     Behavior likelihood $p(y|\mathbf{s}_t) \leftarrow w_t \odot \mathcal{A}_t - \rho \times \text{item\_correlation}(\mathcal{A}_t)$
5:     Sample final immediate feedback $\mathcal{Y}_t^{(I)} \sim p(y|\mathbf{s}_t)$

**The user leave module:**

6:     Immediate reward $r_t \leftarrow \text{reward\_func}(\mathcal{Y}_t^{(I)})$
7:     User temper $\leftarrow$ user temper - immediate reward
8:     Leave signal $\mathcal{Y}_t^{(L)} \leftarrow 1$ if user temper $\leq \mathbb{T}$; 0 otherwise

**The user retention module:**

9:     Personal retention bias $b_u \leftarrow \text{DNN}(\mathbf{s}_t)$
10:    Response retention bias $b_r \leftarrow \lambda_1 r_t$
11:    Next day return probability $p_{\text{ret}} \leftarrow b_u + b_r + \lambda_2 b$, where $b$ is the global retention bias
12:    Return time $\mathcal{Y}_t^{(R)} \sim \text{Geometric}(p_{\text{ret}})$ if $\mathcal{Y}_t^{(L)} = 1$, otherwise $\mathcal{Y}_t^{(R)} = 0$

**Post processing module:**

13:    Update user history $\mathcal{H}_{:t} \leftarrow \mathcal{H}_{:t-1} \oplus (\mathcal{A}_t, \mathcal{Y}_t^{(I)}, \mathcal{Y}_t^{(L)}, \mathcal{Y}_t^{(R)})$
14:    If $\mathcal{Y}_t^{(L)} == 1$, the user leave the current session
15:    Else if not reaching the max session number, then continue.
16:    Otherwise, sample a new user $\mathcal{U}, \mathcal{H}$ from data and replace the current user.

---

**Request level** This problem focuses on single request optimization where the recommendation list could be optimized in a list-wise way. Different from the standard learning-to-rank paradigm where a point-wise model is trained to predict the ranking score of each individual item, the list-wise recommendation assumes that each recommendation action corresponds to a list of items that are exposed to the user in order, and it addresses the existence of item correlations within recommendation lists [29]. In this case, conventional point-wise evaluations like Recall and NDCG metrics become ineffective in evaluating these cross-item patterns, and simulators are introduced to fill in the gap.

**Whole-session level** When we look at each user session as a whole and want to optimize the overall recommendation performance of the entire session, cross-request influences are included in the picture and the problem corresponds to the most standard MDP setting for RL-based models: Each recommendation drives a user state transition and thus the long-term performance in the session, which could be effectively modeled as a Markov Decision Process [1].

**Cross-session level** Further expanding our view to the entire interaction sequences across sessions, an extra user return time signal becomes observable, which refers to the time interval between the last request of a session and the first request of the subsequent session. This introduces an additional learning goal of retention optimization, which aims to learn a policy that provides satisfactory recommendations that can reduce the user's return time. Note that this learning goal is directly related to daily average users (DAU) which is the core evaluation metric for many web services. However, its optimization is extremely challenging for its uncertainty and uncontrollable delay [3].

### 3.2 Simulator Building Blocks

As we would simulate the online environment and provide a training and evaluation platform for all aforementioned tasks, our simulator aims to provide the multi-behavior immediate responses, the user leave model, and the user retention model. Formally, the user simulator is a response generation function for recommendation actions $\mathcal{S} : \mathcal{O}_t, \mathcal{A}_t \rightarrow \mathcal{Y}_t$. We provide a detailed workflow of this function in Algorithm 1 which consists of three feedback generation modules:

**User immediate response module (UIRM)** is responsible for generating the user's immediate feedback $\mathcal{Y}_t^{(I)}$. It first infers a ground-truth user state $\mathbf{s}_t$ (assumed implicitly from RL models), then outputs the behavioral likelihood for each immediate feedback type. Specifically, $\oplus$ represents concatenation and $\odot$ represents dot product. The item\_correlation function is introduced to suppress the positive behaviors for items with higher correlation with other items in the same recommendation

Table 2: Statistics of datasets.

| Datasets | Users | Items | Interactions | Sessions | Density |
|---|---|---|---|---|---|
| KuaiRand-Pure | 27077 | 7551 | 1,436,609 | 246738 | 0.70% |
| ML-1m | 6,400 | 3,706 | 1,000,208 | 16629 | 4.22% |

list. This behavior simulates the user's demands for item diversity since lower item correlation induces a higher chance of positive interactions. In order to ensure the validity of the UIRM model, we require pretraining on the log data, and for each immediate feedback type, we can adapt the standard point-wise learning using binary cross-entropy. As a result, any datasets that provide logs of user-wise recommendation-feedback sequences would sufficiently support this module.

**User leave module** maintains a user temper/patience factor that directly determines the leave signal. We assume a maximum length of each user session, and initialize the user temper as equal to this maximum length. Then the user gradually loses the temper during the interactions and finally leaves the session when the temper is too low. In each step, the $\mathcal{Y}^{(I)}$ inferred by the UIRM is reused to calculate the immediate reward that represents the user's satisfaction with the $\mathcal{A}_t$. We assume that less satisfactory recommendations make users lose their temper faster. The initial temper value, the decrease rate of temper, and the leave threshold are all adjustable hyperparameters.

**User retention module** is specifically designed for cross-session tasks, which predicts the user's return time. The user's return time typically follows a geometric distribution (as we will further discuss in section 3.3), so we only predict a next-day return probability $p_{\text{ret}}$ to simulate this behavior. Specifically, $p_{\text{ret}}$ combines the global retention bias, a personal retention bias, and a response retention bias. The personal retention bias reflects the differences in user's activity level, e.g., highly active users may use the system every day with next-day return probability, but lower-activity users may come back to the system after several weeks. The response retention bias assumes that better recommendations also increase the user's returning probability since users are more satisfactory to the system. We tune the global bias to fit the geometric distribution of the data and restrict the return day signal in $\mathcal{Y}_t^{(R)} \in \{1, \ldots, D\}$ after each session. Here, we set $D = 10$ in our simulation because the percentage of requests returned after the tenth day is almost zero.

### 3.3 Dataset Analysis

In order to maintain distributional consistency between the simulator and the real-world environment, we build KuaiSim with the support of log data. The data will be used to pretrain the UIRM and sample the user's initial observation during online interactions as the post-processing module in algorithm 1. In this section, we further describe the example datasets used in building up the simulators.

**Dataset description** We construct our simulator KuaiSim on the KuaiRand [10] dataset, a large-scale unbiased sequential recommendation dataset collected from the Kuaishou[4] App. It elicits user preferences on the randomly exposed items to collect unbiased data. You can find detailed data collection process in Appendix A.1. The availability of unbiased sequential data allows us to conduct unbiased offline evaluations, which greatly aids in constructing our simulator. We specifically choose the KuaiRand-Pure version. By doing so, we maintain a focused dataset for constructing our simulator. The KuaiRand dataset encompasses 12 distinct feedback signals. We utilize six positive feedback signals (i.e., 'click', 'view time', 'like', 'comment', 'follow', and 'forward'). Additionally, we incorporate two negative feedback signals (i.e., 'hate' and 'leave'). The other feedback signals occur infrequently and are therefore excluded from our analysis. Furthermore, we broaden the scope of our simulator by incorporating the ML-1m[5] dataset. This dataset serves as a benchmark commonly used in RSs and includes ratings provided by 6,014 users for 3,417 movies. In our analysis of the ML-1m dataset, we consider movies with user ratings higher than 3 as positive samples, representing a 'like,' while the rest are deemed negative samples, representing a 'hate.'

**Dataset process** We standardize both datasets by preprocessing them into a cohesive format, ensuring that each record follows a sequential order of (user features, user history, exposed items, user feedback, and timestamp). The detailed statistics of the resulting dataset can be found in Table 2. To guarantee the data quality of the KuaiRand dataset, we apply 50-core filtering where videos with less than 50 occurrences are removed. To generate session data, we segment each day chronologically, treating each day as an individual session.

---

[4] https://www.kuaishou.com/cn
[5] https://grouplens.org/datasets/movielens/1m/

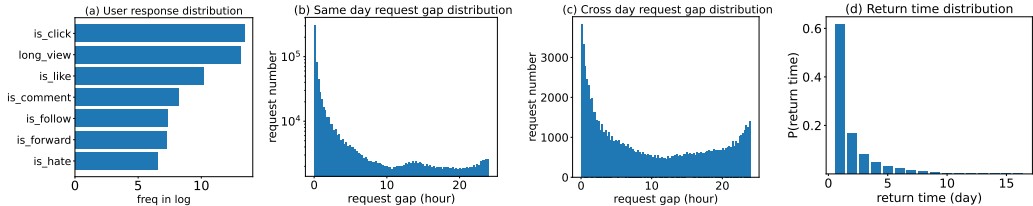

Figure 2: Data analysis on KuaiRand dataset. (a) User response distribution. (b) Same day request gap distribution. (c) Cross day request time gap distribution. (d) Return time distribution

Table 3: Benchmarks for the request level task of KuaiSim. Best values are in bold.

| Algorithm | Average L-reward | Max L-reward | Coverage | ILD |
|-----------|------------------|--------------|----------|-----|
| CF | **2.253 ± 0.024** | 4.039 ± 0.001 | 100.969 ± 7.193 | 0.543 ± 0.007 |
| ListCVAE | 2.075 ± 0.039 | **4.042 ± 0.001** | **446.100 ± 15.648** | **0.565 ± 0.004** |
| PRM | 2.174 ± 0.017 | 3.811 ± 0.003 | 27.520 ± 3.210 | 0.538 ± 0.004 |

**Primary data analysis** We present our primary data analysis on the KuaiRand-Pure dataset in Figure 2. A full analysis procedure and more detailed results can be found in Appendix A.2. In our observation: 1) Figure 2(a) displays the distribution of user feedback, showcasing the frequency of occurrences for the six positive and one negative ('hate') 0/1 feedback signals. Notably, the negative feedback signal, 'is hate,' appears the least frequently. Among the positive feedback signals, 'is click' exhibits the highest frequency, while 'is forward' has the lowest occurrence. 2) In Figure 2(b) and 2(c), we present the distribution of same day and cross day request gaps, respectively. These gaps represent the time intervals, in hours, between consecutive requests. The occurrences of different request time gaps follow a geometric distribution pattern. 3) By analyzing the frequency of session returns and plotting the return time distribution in Figure 2(d), we observe that the return time also adheres to a geometric distribution, i.e., given $p_{\text{ret}}$, the $d$-th day return probability is $(1 - p_{\text{ret}})^{d-1}p_{\text{ret}}$. And note that the percentage of return time greater than 10 is nearly negligible.

## 4 Baselines and Evaluation Protocol

### 4.1 Baseline Methods

**List-wise recommendation with request level simulator**: For list-wise recommendation, we first consider a standard collaborative filtering baseline **CF** [17]. It adopts a point-wise approach, where the user-item scoring function is trained through binary cross entropy. We then include a generative approach **ListCVAE** [16], which captures the distribution of recommended lists using a conditional VAE. We also include a reranking approach **PRM** [36], which is built upon a CF-based initial ranker and utilizes a transformer-based re-ranker to encode the intermediate candidate set.

**Sequential recommendation with whole-session simulator**: With the standard MDP formulation, we include the following RL methods as comparisons: **A2C** [31] is a synchronized version of A3C [31] that applies the policy gradient algorithm on the actor and TD error learning on the critic. **SA2C** [50] combines a negative sampling strategy with supervised sequential learning to enhance reward signal. **DDPG** [26] is a deep deterministic policy gradient algorithm and the continuous action space is used to represent the recommendation list, for both the actors and critics. **TD3** [9] is an extension of DDPG that improves the stability of training by incorporating the clipped double Q-learning. **HAC** [28] is also an extension of DDPG that decomposes the generation of item lists into a hyper-action(embedding) inference step and an effect-action(item list) selection step.

**Retention optimization with cross-session simulator**: The retention optimization task is essentially a hyper-parameter search problem. We first consider **CEM** [39] as a black-box optimization method. It iteratively samples and updates a population of candidate hyper-parameters based on their performance. We then include **TD3** [9] method similar to that in whole-session level, but use return time as the reward. We also include **RLUR** [3] as the state-of-the-art method. It uses intrinsic rewards [34] to enhance policy learning and introduces a novel soft regularization method to address the trade-off between sample efficiency and stability.

Table 4: Benchmarks for the whole-session task of KuaiSim. Best values are in bold.

| Algorithm | Depth | Average reward | Total reward | Coverage | ILD |
|---|---|---|---|---|---|
| TD3 | $14.63 \pm 0.03$ | $0.6476 \pm 0.0028$ | $9.4326 \pm 0.0756$ | $24.20 \pm 2.55$ | $0.9864 \pm 0.0004$ |
| A2C | $14.02 \pm 0.02$ | $0.5950 \pm 0.0019$ | $8.3905 \pm 0.1026$ | $27.41 \pm 1.08$ | $0.9870 \pm 0.0002$ |
| SA2C | $14.34 \pm 0.02$ | $0.6251 \pm 0.0014$ | $8.9547 \pm 0.0241$ | $27.14 \pm 2.01$ | $0.9872 \pm 0.0002$ |
| DDPG | $14.89 \pm 0.04$ | $0.6841 \pm 0.0013$ | $10.0873 \pm 0.0571$ | $20.95 \pm 3.27$ | $0.9850 \pm 0.0006$ |
| HAC | $\mathbf{14.98 \pm 0.03}$ | $\mathbf{0.6895 \pm 0.0017}$ | $\mathbf{10.1742 \pm 0.0634}$ | $\mathbf{35.70 \pm 1.22}$ | $\mathbf{0.9874 \pm 0.0004}$ |

Table 5: Benchmarks for the cross-session task of KuaiSim. Best values are in bold.

| Algorithm | Return day $\downarrow$ | User retention $\uparrow$ |
|---|---|---|
| CEM | $3.573 \pm 0.012$ | $0.572 \pm 0.002$ |
| TD3 | $3.556 \pm 0.010$ | $0.581 \pm 0.001$ |
| RLUR | $\mathbf{3.481 \pm 0.010}$ | $\mathbf{0.607 \pm 0.002}$ |

$\uparrow$: the higher the better; $\downarrow$: the lower the better.

## 4.2 Evaluation Protocol

**List-wise recommendation with request level simulator**: **List-wise reward (L-reward)** is the average of item-wise immediate reward. We use both the average L-reward and the max L-reward across user requests in a mini-batch. **Coverage** describes the number of distinct items exposed in a mini-batch. **Intra-list diversity (ILD)** estimates the dissimilarity between items in each recommended list, based on item embedding.

**Sequential recommendation with whole-session simulator**: Besides coverage and ILD, we also use other metrics. **Whole-session reward**: total reward is the average sum of immediate rewards for each session. The average reward is the average of total reward for each request. **Depth** represents how many interactions before the user leaves.

**Retention optimization with cross-session simulator**: **Return day** is the average time gap (day) between the last request of session and the first request of session. **User retention** is the average ratio of visiting the system again.

## 5 Benchmark Results and Analysis

We provide benchmark results for all three tasks in this section. To ensure the reproducibility and reliability of results, we computer averages across five distinct sets of results. The $\pm$ represents the standard deviation of five results. Detailed experiment implementation and hyper-parameter settings are provided in Appendix B.

**List-wise recommendation with request level simulator**: As shown in Table 3, among the evaluated models, ListCVAE demonstrates the best performance in terms of maximum reward and diversity. Its ability to generate diverse and high-reward recommendations makes it an effective choice for list-wise recommendation tasks. On the other hand, PRM exhibits the worst performance among the evaluated models. A significant challenge in this task lies in efficiently searching the extensive combinatorial space of list actions. A promising avenue for future research involves addressing this challenge by simultaneously improving the diversity of recommendation results and reducing the complexity of the combination space.

**Sequential recommendation with whole-session simulator**: As shown in Table 4, the HAC framework consistently demonstrates superior performance across long-term metrics, showcasing the effectiveness of its hyper-actions and the efficiency of its learning method. Notably, HAC outperforms other frameworks, indicating its high level of expressiveness and efficacy in recommendation tasks. On the other hand, A2C exhibits the poorest performance and appears to be the most unstable learning framework among the evaluated methods. SA2C outperforms A2C by capitalizing on an improved reward signal generated through negative sampling and supervised training. While slightly trailing behind HAC, the DDPG framework also delivers commendable results. Current methods in this task often overlook the importance of incorporating long-term feedback. Therefore, a promising research direction lies in exploring how to model complex inter-session relationships effectively.

Table 6: A comparison between KuaiSim and other simulators. Best values are in bold.

| Simulators | Depth | Average reward | Total reward | AUC |
|---|---|---|---|---|
| RL4RS | $14.39 \pm 0.02$ | $0.640 \pm 0.015$ | $9.235 \pm 0.122$ | $0.6929 \pm 0.0019$ |
| Recogym | $13.55 \pm 0.01$ | $0.535 \pm 0.013$ | $7.194 \pm 0.109$ | $0.6729 \pm 0.0026$ |
| RecSim | $14.05 \pm 0.02$ | $0.588 \pm 0.006$ | $9.347 \pm 0.143$ | $0.6842 \pm 0.0031$ |
| VirtualTaobao | $14.45 \pm 0.02$ | $0.646 \pm 0.009$ | $9.570 \pm 0.077$ | $0.6866 \pm 0.0014$ |
| KuaiSim | $\mathbf{14.86^* \pm 0.01}$ | $\mathbf{0.679^* \pm 0.011}$ | $\mathbf{10.081^* \pm 0.116}$ | $\mathbf{0.7234^* \pm 0.0021}$ |

"**\***" indicates the statistically significant improvements (i.e., two-sided t-test with $p < 0.05$) over the best baseline.

Table 7: Benchmarks for the whole-session task on ML-1m datasets. Best values are in bold.

| Algorithm | Depth | Average Reward | Total reward | Coverage | ILD |
|---|---|---|---|---|---|
| TD3 | $13.50 \pm 0.01$ | $0.5388 \pm 0.007$ | $7.4035 \pm 0.0152$ | $44.18 \pm 2.19$ | $0.9866 \pm 0.0001$ |
| A2C | $13.55 \pm 0.01$ | $0.5468 \pm 0.002$ | $7.4487 \pm 0.0171$ | $27.31 \pm 1.44$ | $0.9870 \pm 0.0001$ |
| DDPG | $13.64 \pm 0.02$ | $0.5476 \pm 0.005$ | $7.5333 \pm 0.0273$ | $\mathbf{61.03 \pm 2.08}$ | $0.9871 \pm 0.0001$ |
| HAC | $\mathbf{13.70 \pm 0.01}$ | $\mathbf{0.5482 \pm 0.008}$ | $\mathbf{7.5791 \pm 0.0206}$ | $29.85 \pm 1.92$ | $\mathbf{0.9872 \pm 0.0001}$ |

**Retention optimization with cross-session simulator**: As shown in Table 5, TD3 demonstrates better performance than CEM in terms of both metrics, showcasing the effectiveness of reinforcement learning techniques. However, RLUR surpasses both TD3 and CEM significantly, indicating its superior performance in the evaluated tasks. Indeed, the exploration of this task is still in its early stages. While some research has attempted to model user retention on a day-by-day basis, capturing more enduring user feedback has considerable potential in exploring approaches.

## 6 Discussion

**Comparing with other simulators** In section 2.4, we compare our KuaiSim with other simulators qualitatively in terms of datasets and tasks, showing that our KuaiSim is the only one that meets all requirements. In this section, we compare our KuaiSim with other simulators quantitatively. We reconstruct some competitive simulators on the KuaiRand dataset for the whole-session task to illustrate the effectiveness of our KuaiSim. In order to evaluate the simulator, we divide it into two planes of evaluation. On the one hand, how accurately the simulator fits the environment. Since some simulators have only one feedback signal, click, we compare the AUC predicted for this signal. On the other hand, the agent is trained using the simulator. We utilize three metrics proposed in the whole-session evaluation protocol, depth, average reward, and total reward. As shown in Table 6, we utilize the DDPG algorithm to train an agent with different simulators. Among these simulators, KuaiSim outperforms the others by a significant margin across all evaluation metrics. This outcome demonstrates that KuaiSim accurately aligns with the environment, resulting in superior agent training. Compared to rule-based simulators like RecSim and RecoGym, our simulator can be trained in a supervised manner to fit the real environment better. Additionally, compared to Virtual-Taobao and RL4RS, KuaiSim directly samples users from datasets without the need for fitting user states, avoiding potential errors. These distinctions highlight the strengths of our simulator, KuaiSim, which contributes to its enhanced accuracy and reliability in simulating user behavior and preferences.

**Data migration analysis** To showcase the data migration capabilities of our construction process, we implement KuaiSim on the ML-1m dataset and evaluate its performance on the whole session task. The benchmark results, as presented in Table 7, indicate that HAC continues to outperform other methods in all metrics, except for coverage. Notably, DDPG exhibits the highest coverage and achieves the best diversity in the recommended results. On the other hand, A2C outperforms TD3, positioning itself as a stronger-performing model compared to the latter. TD3 exhibits the poorest performance among the evaluated models. In a nutshell, these findings show that our KuaiSim can work well on ML-1m and emphasize the effectiveness of KuaiSim in adapting to different datasets.

**Parameter analysis** We further investigate the effect of some vital hyper-parameters for training the Algorithm TD3 with our KuaiSim. We examine the influence of max time step in the range of [5, 10, 15, 20, 25, 30], the results are presented in Figure 3 (a). Optimal model performance is achieved when the max time step is set to 20, and the return day fluctuates between 2.2 and 2.3 for other settings. Additionally, we delve into the effect of slate size, varying it across [5, 10, 15, 20,

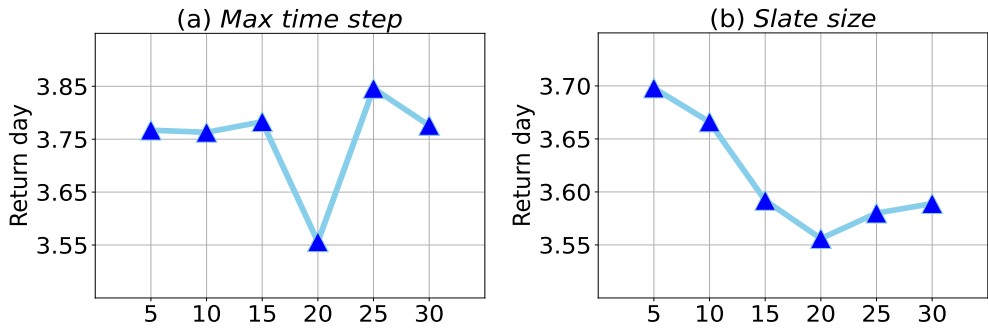

Figure 3: Parameter sensitive analysis on KuaiRand dataset. (a) Max time step. (b) Slate size.

25, 30]. the results are shown in Figure 3 (b). The best model results are obtained when the slate size is set to 20. The performance exhibits a decline as the slate size deviated from this optimum point, either increasing or decreasing. This is because when the slate size is excessively small, the recommendation list fails to encompass items that align with the user's preferences. Conversely, when the slate size is overly large, the recommendation list contains too much noise that is not of interest to the user. This overload of information not only diverts the user's focus but also tests their patience. The model's outcomes demonstrate a consistent and stable pattern across different parameter settings, underscoring the robust stability of our KuaiSim training agent.

**Limitations and potential direction for solutions** There remain certain areas within our RecSim framework that warrant refinement. **Uncovered recommendation problems:** While we propose three levels of recommendation tasks to encompass a majority of recommendation challenges, numerous others exist, including explainable recommender systems [6, 33], diversity recommendations [13, 19], and fairness recommendations [47, 23]. Incorporating the interpretability, diversity, fairness, and other indicators of the recommendation results into the modeling of the simulator is a promising strategy to broaden our simulator's scope. **Expanding to diverse domains:** While both the KuaiRand and ML-1m datasets center around video recommendations, it's worth noting that the KuaiSim framework can be extended to encompass a variety of domains such as music, news, and e-commerce. The iterative refinement process we've undertaken for constructing and evaluating the simulator greatly facilitates such expansions.

**Ethical considerations and potential societal impact** We categorize our ethical considerations and potential societal impact into three key aspects. **Safeguarding user privacy:** While our approach involves training the simulator using real user logs, it's important to note that the datasets we employed have been stripped of any privacy-sensitive information. The user characteristics utilized in our study are comprehensively presented in Appendix A.1 Table 2. These characteristics reflect the user's interactions with the application, and each user is denoted by an anonymous uid. Thus, it's essential to emphasize that we do not leverage any private user information, effectively mitigating concerns related to privacy breaches. **Unbiased KuaiRand dataset:** Because online A/B test usually consumes much time and money, which makes it impractical for academic researchers to conduct the evaluation online. However, offline evaluation will cause bias because of massive missing data, i.e., the user-item pairs that have not occurred in the test set. One fundamental approach to address this issue in offline evaluation is to collect unbiased data, i.e., to elicit user preferences on the randomly exposed items. This unbiased dataset empowers us to conduct unbiased offline evaluations without compromising user privacy. The detailed data collection process can be found in Appendix A.1. **Promoting field advancement:** We provide various competitive algorithms for three task levels as benchmarks, coupled with a meticulously enhanced process for constructing and evaluating simulators. This framework not only facilitates the advancement of the field but also presents a promising avenue for research by extending these simulators to encompass a broader spectrum of tasks and fields.

# 7 Conclusion

In summary, KuaiSim stands as a comprehensive and sophisticated simulator that encompasses multiple task levels, establishing benchmarks and enabling thorough evaluations in the realm of recommendation systems. Through its refined construction and evaluation process, as well as its effectiveness in replicating user behaviors, KuaiSim contributes to the development and advancement of recommendation system techniques and methodologies.

## ACKNOWLEDGEMENT

This research was supported by Research Impact Fund (No. R1015-23). It was also supported by Kuaishou. It was also partially supported by APRC - CityU New Research Initiatives (No.9610565, Start-up Grant for New Faculty of City University of Hong Kong), CityU - HKIDS Early Career Research Grant (No.9360163), Hong Kong ITC Innovation and Technology Fund Midstream Research Programme for Universities Project (No.ITS/034/22MS), Hong Kong Environmental and Conservation Fund (No. 88/2022), and SIRG - CityU Strategic Interdisciplinary Research Grant (No.7020046, No.7020074).

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

# A  Detailed Dataset Description and Analysis

## A.1  Detailed Feature Description

Table 1: Detailed interaction information in KuaiRand dataset.

| Interaction field | Feature type | Explanation |
|---|---|---|
| user id | int64 | The unique identifier for the user. |
| video id | int64 | The unique identifier for the video. |
| date | int64 | The date when the interaction occurred. |
| hour min | int64 | The time of the interaction in hours and minutes. |
| time ms | int64 | The timestamp of the interaction in milliseconds. |
| is click | int64 | The user feedback of click. |
| is like | int64 | Indicating if the user liked the video. |
| is follow | int64 | Indicating if the user followed the author. |
| is comment | int64 | Indicating if the user wrote a comment. |
| is forward | int64 | A binary signal indicating if the user forwarded the video. |
| is hate | int64 | A binary signal indicating if the user disliked the video. |
| long view | int64 | Indicating the completeness of the video. |

Table 2: Rich user feature in KuaiRand dataset.

| User feature | Feature type | Explanation |
|---|---|---|
| user id | int64 | The unique identifier for the user. |
| user active degree | str | The level of user activity, classified as 'high active', 'full active', 'middle active', or 'UNKNOWN'. |
| is live streamer | int64 | Indicates whether the user is a live streamer. |
| is video author | int64 | Indicates if the user has uploaded any videos. |
| follow user num range | str | The number range of followed users. |
| fans user num range | str | The number range of fans. |
| friend user num range | str | The number range of friends. |
| register days range | str | The range of the number of days since user registration. |
| one-hot features | int64 | Encrypted features of the user. |

Table 3: Rich video feature in KuaiRand dataset.

| User feature | Feature type | Explanation |
|---|---|---|
| video id | int64 | The unique identifier for the video. |
| author id | int64 | The unique identifier for the author of the video. |
| video type | str | The type of the video, categorized as "NORMAL" or "AD". |
| upload type | str | The upload type of the video. |
| music type | int64 | The background music type used in the video. |
| tag | str | A list of key categories or labels associated with the video. |

This section offers an elaborate overview of the KuaiRand [10] dataset, presenting its key components in detail. Because online A/B test usually consumes much time and money, which makes it impractical for academic researchers to conduct the evaluation online. However, offline evaluation will cause bias because of massive missing data, i.e., the user-item pairs that have not occurred in the test set. A way to fundamentally solve this problem in offline evaluation is to collect unbiased data, i.e., to elicit user preferences on the randomly exposed items. To achieve this goal, sample a batch of videos and filter out the spam such as advertisements. There are 7,583 items in total. For the target users, randomly select a batch of users and remove robots, which includes over 200,000 real users. Each time the recommender system recommends a video list to a user, decide whether to insert a random item with a fixed probability. If the answer is yes, then intervene in the recommendation list by randomly selecting one video from this list and replacing it with a random item uniformly sampled from the 7,583 items. KuaiRand removes the users that have been exposed to less than 10 randomly exposed videos for faithful evaluation. There are 27,285 users retained. All 7,583 items have been inserted at least once, and the total number of random interventions is 1,186,059. The KuaiRand dataset

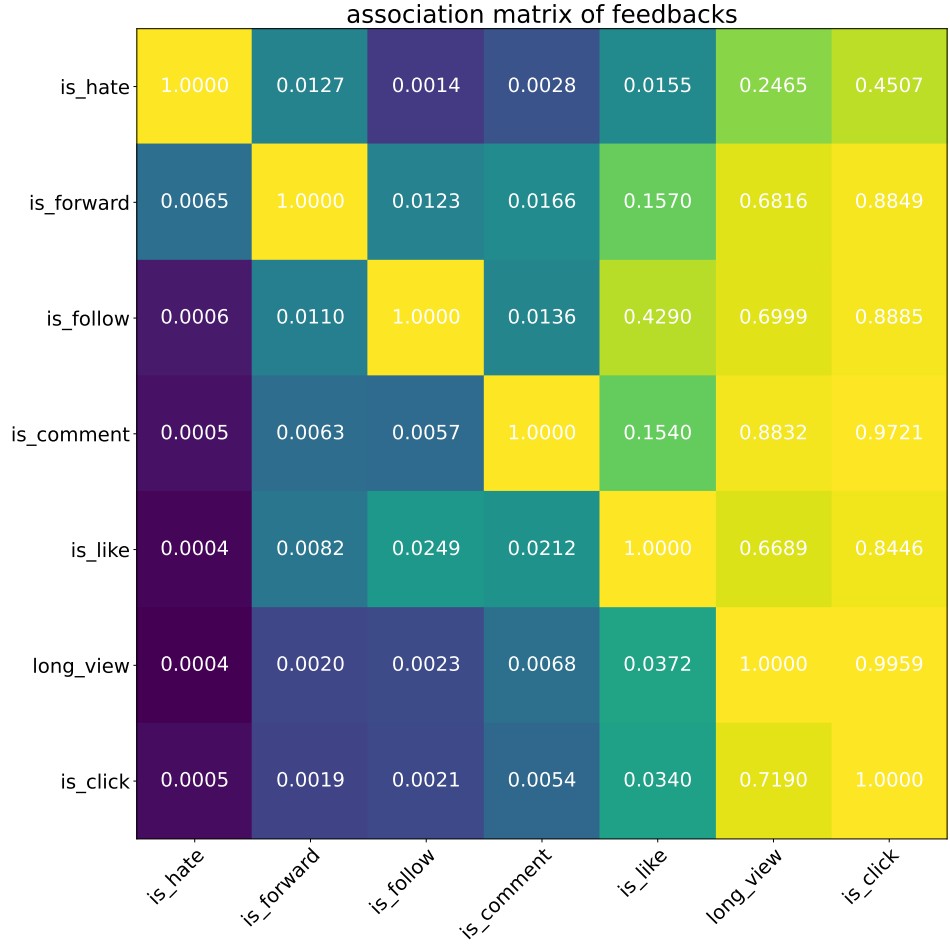

Figure 1: Association matrix of seven kinds of feedback.

encompasses five distinct components: an interaction dataset, a dataset comprising rich user features, and a dataset consisting of rich video features. We provide a detailed compilation of these components we used, including the names, feature types, and explanations, which are presented in Table 1 for interaction features, Table 2 for user features, and Table 3 for item features. We have meticulously selected high-quality features for both the user and item categories. From the 17 available one-hot features for the user, we have selected the following six features: [0, 1, 6, 9, 10, 11]. These specific features have been chosen to provide meaningful insights and additional dimensions for user analysis within the dataset. These features have been thoughtfully curated to ensure their relevance, reliability, and usefulness in gaining meaningful insights and understanding the user dynamics within the dataset. For detailed information about all the features included in the KuaiRand dataset, you can refer to the main page of the dataset [6].

### A.2 Further data analysis

Figure 1 presents a heatmap that illustrates the association matrix of seven different types of feedback. The heatmap provides a visual representation of the relationships between these feedback categories. When users provide negative feedback, such as expressing hate towards a video, they tend to give very few other positive feedback signals. On the other hand, among the positive feedback signals, both clicks and long views exhibit a strong association with other positive feedback indicators. Conversely, forwarding a video shows the least association with other positive feedback signals. These observations highlight distinct patterns in user behavior when it comes to expressing negative feedback and engaging in various positive feedback actions.

---

[6]https://kuairand.com/

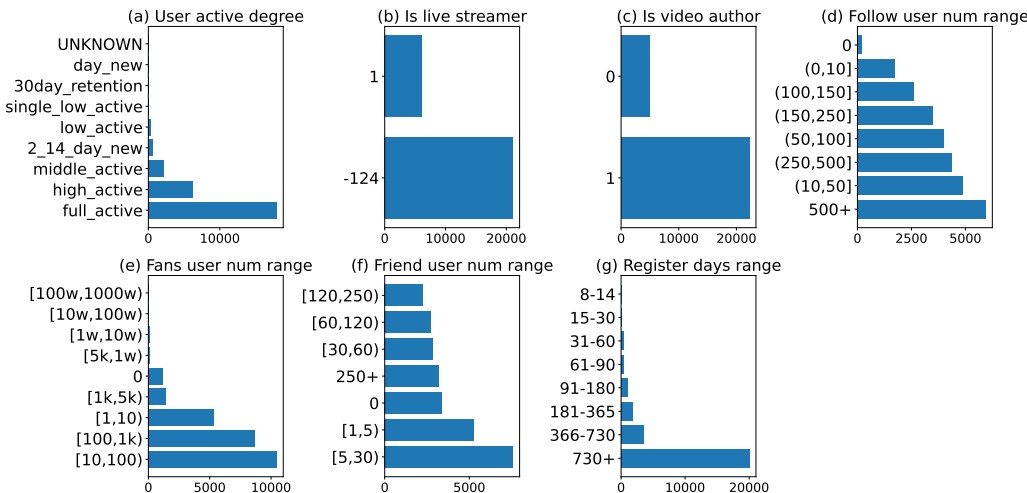

Figure 2: Rich user feature distribution. (a) User active degree. (b) Is live streamer. (c) Is video author. (d) Follow user num range. (e) Fans user num range. (f) Friend user num range. (g) Register days range.

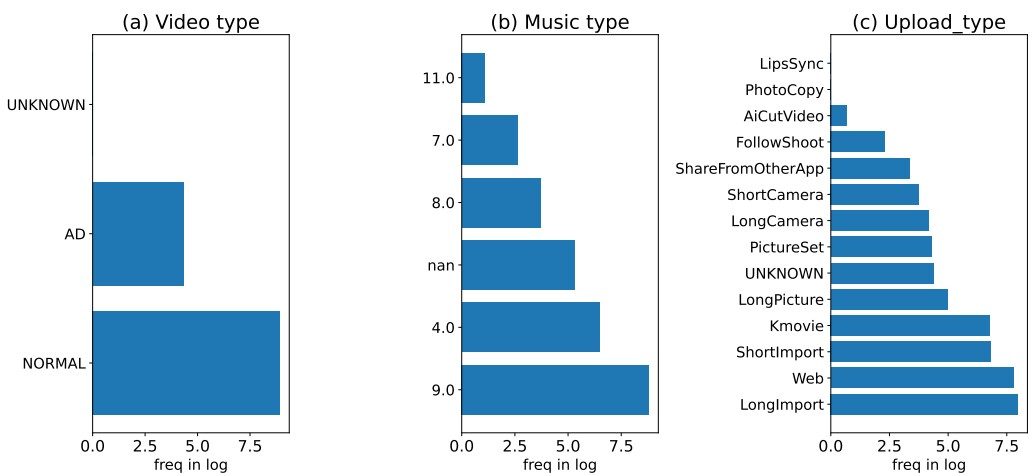

Figure 3: Rich video feature distribution. (a) Video type. (b) Music type. (c) Upload type.

Furthermore, Figure 2 provides valuable insights into the distribution of the user features utilized in our model. Regarding the feature 'user active degree', the majority of users are classified as 'full active', followed by 'high active', while the remaining categories represent a significantly smaller proportion. In terms of the 'live streamer' feature, the vast majority of users (-124) are not live streamers. However, when considering the 'video author' feature, a large majority of users are indeed video authors. Examining the 'follow user num range' feature, the category '500+' dominates the distribution, while the category '0' represents a minimal proportion. Analyzing the 'fans user num range' feature, the range '[10, 100]' captures the largest share, with both '[1, 10]' and '[100, 1k]' accounting for substantial percentages, and the remaining ranges having little to no representation. For the 'friend user num range' feature, the majority of users fall within the range '[5, 30]'. Lastly, the 'register days range' feature shows that the category '[730+]' encompasses nearly all users, indicating a significant proportion of long-standing registered accounts. This visual representation allows for a comprehensive understanding of the characteristics and patterns present within the user data.

Similarly, Figure 3 visually represents the distribution of video features employed in our model. Examining the 'video type' feature, the majority of videos fall under the category of 'NORMAL', while the 'UNKNOWN' category represents a significantly smaller proportion. Regarding the "music type" feature, the value '9.0' dominates the distribution, indicating a prevalent use of a specific

background music type. Additionally, a substantial proportion of videos have no background music at all. Analyzing the 'upload type' feature, the category 'LongImport' captures the largest share, while the categories 'LipsSync' and 'PhotoCopy' account for a significantly smaller proportion. This visualization aids in the exploration of the video dataset, enabling the identification of noteworthy trends or peculiarities within the data. These figures collectively enhance our understanding of the underlying patterns, associations, and distributions within the datasets, thereby facilitating a more robust and insightful model.

## B  Detailed Experiment Implementations

To ensure simplicity and consistency, we have established certain parameter settings and search spaces for our experiments. We set the embedding size of $\mathcal{U}$ and $\mathcal{H}_{:t-1}$ from [32, 64, 128]. The latent embedding size is two times the input dimension. In the immediate response module, we assign an immediate reward weight of 1 to all immediate feedback signals except for the hate signal, which is assigned a weight of -1. Consequently, the range of immediate reward, denoted as $r$, spans from -1 to 6 in the KuaiRand dataset, and from -1 to 2 in the ML-1m dataset. The layer number of DNN and Transformer is set to 2. The multi-head of the Transformer is set to 2. The dropout rate is set to 0.2. In the user leave module, we set the max time step and initial temper value from [5, 10, 15, 20, 25, 30], and the rate of temper decrease as 1. Additionally, we define the leave threshold as 1. Furthermore, in the user retention module, $\lambda_1$ is set to 0.5, and $\lambda_2$ is set to 0.75. The number of DNN layers is set to 2. The slate size is from [5, 10, 15, 20, 25, 30]. For the actor learning rate, we explore a common search space consisting of [0.0005, 0.0001, 0.00005, 0.00001, 0.000005, 0.000001]. Similarly, we investigate the critic learning rate in the range of [0.001, 0.0001, 0.00001]. For both simulator and agent training, the batch size is set to 64. And we search for the optimal learning rate within the range of [0.0005, 0.0001, 0.00005, 0.00001]. Moreover, we perform L2 regularization with coefficients selected from [0.0001, 0.00005, 0.00001, 0.000005]. When comparing to other baselines, we either utilize the same search range or adopt the optimal settings recommended by the original authors of the baselines. We divide the dataset as training set and test set with a ratio of 8:2, which is a common setting in previous works. All experiments are conducted on an NVIDIA Tesla V100S GPU, and the reported results are the average of three replicate experiments.

**Reproduction guidelines** Below we will illustrate how to reproduce our experimental results. Please use Python 3.8 (or later), torch 2.0.0 (or later). If you want to use GPU acceleration, please use CUDA 11.8 (or later). Then you can follow the guidelines below:

- Install the necessary packages according to our '0.Setup'.
- Find the details of data preprocessing in 'code/preprocess/KuaiRandDataset.ipynb'.
- Run 'run_nultibehavior.sh' to train the simulator.
- Run 'generate_session_data.sh' to generate session data for agent training.
- Then you can run the 'train_(agent name)_krpure_(task level).sh' file to train different agents on different tasks with our simulator.
- We provide visualized training results in 'TrainingObservation.ipynb'.

We have provided detailed parameter settings in all the sh files. For more analysis experiments, you can also check out the project READ ME. And we will be working on further improvements to our project, such as readability guarantees for existing code and the incorporation of novel baselines.

