# OpenReview forum: "KuaiSim: A Comprehensive Simulator for Recommender Systems"
_NeurIPS.cc/2023/Track/Datasets_and_Benchmarks — NeurIPS 2023 Datasets and Benchmarks Poster_

### Official Review · Reviewer_P5ds · 2023-07-20
**The simulator helps to produce datasets that are more consistent with real-world data.**

**Rating:** 7
**Confidence:** 2
**Correctness:** Yes
**Clarity:** Yes

**Strengths:**

1. The proposed simulator is comprehensive;
2. This paper is well-written;
3. The implementation code is available.


**Additional Feedback:**

NA

**Documentation:**

Yes

**Limitations:**

I'm not quite sure

(1) how the authors evaluated whether a simulator was better or worse,

(2) how they evaluated the simulator as being more consistent with real-world data.

(3) Also, in the experiments, is there a split between the training set, validation set, and test set?

**Opportunities For Improvement:**

1. it would be better to conduct some sensitivity analysis;

2. No checklist available

**Relation To Prior Work:**

Yes

**Summary And Contributions:**


This paper introduces a comprehensive simulator called KuaiSim, which offers users feedback with multi-behavior and cross-session responses. KuaiSim is designed to handle three levels of recommendation problems: the request-level list-wise recommendation task, the whole-session level sequential recommendation task, and the cross-session level retention optimization task.

To evaluate the performance and behavioral differences, the authors compare KuaiSim against existing competitive simulators on the Kuairand Dataset. Additionally, a comparative analysis is conducted to demonstrate that KuaiSim outperforms other simulators in its ability to accurately simulate real-world environments. The implementation code is available online to ease reproducibility.

---

> ### Author Response · Authors · 2023-08-18
> **Feedback to Reviewer #3**
>
> We appreciate your overall positive assessment of our contributions and are grateful for your suggestion. We have already updated the new revision based on your suggestions.
>
> ------
>
> **Q1: Sensitivity analysis**
>
> **A1:** Thanks for your suggestions. We conduct parameter sensitivity analysis for two important parameters max time step and slate size in the range of [5, 10, 15, 20, 25, 30]. The return day results of different parameter settings are shown as follows:
>
> |  Parameters   |   5   |  10   |  15   |    20     |  25   |  30   |
> | :-----------: | :---: | :---: | :---: | :-------: | :---: | :---: |
> | Max time step | 2.220 | 2.216 | 2.236 | **2.009** | 2.299 | 2.229 |
> |  Slate size   | 2.151 | 2.119 | 2.045 | **2.009** | 2.033 | 2.042 |
>
> Optimal model performance is achieved when the max time step is set to 20, and the return day fluctuates between 2.2 and 2.3 for other settings. The best model results are obtained when the slate size is set to 20. And the performance exhibit a decline as the slate size deviated from this optimum point, either increasing or decreasing. This is because when the slate size is excessively small, the recommendation list fails to encompass items that align with the user's preferences. Conversely, when the slate size is overly large, the recommendation list contains too much noise that is not of interest to the user. This overload of information not only diverts the user's focus but also tests their patience. The model's outcomes demonstrate a consistent and stable pattern across different parameter settings, underscoring the robust stability of our KuaiSim training agent.
>
> We plot this result in Figure 3 and a more detailed description is provided in Section 6 in the new revision.
>
> ------
>
> **Q2: No checklist available**
>
> **A2:**  Thanks for your suggestions. We add the checklist after the references section in the new revision. And we check our paper against the checklist and add the following:
>
> - For all authors:
>   - Limitations of our work (See Section 6)
>   - Potential societal impacts (See Section 6)
>   - Ethical considerations (See Section 6)
> - If you ran experiments:
>   - Data splits (See Section 3.3)
> - If you are using existing assets or curating/releasing new assets:
>   - Discuss whether the data contains personally identifiable information (See Section 6)
>
> For dataset splits, we divide the dataset as training set and test set with a ratio of 8:2, which is a common setting in previous works. We make this setting clear in Section 3.3.
>
> ------
>
> **Q3: Simulator evaluate**
>
> **A3:** Thank you for your good questions. We evaluate our simulator both qualitatively and quantitatively.
>
> |   Simulators   | Real dataset | Request-level task | Whole-session task | Cross-session task |
> | :------------: | :----------: | :----------------: | :----------------: | :----------------: |
> |    RecoGym     |              |                    |      &check;       |                    |
> |     RecSim     |              |                    |      &check;       |                    |
> |     RL4RS      |   &check;    |      &check;       |      &check;       |                    |
> | Virtual-Taobao |   &check;    |                    |      &check;       |                    |
> |  **KuaiSim**   |   &check;    |      &check;       |      &check;       |      &check;       |
>
> In Section 2.4 Table 1 as shown above, we compare our KuaiSim with other simulators qualitatively in terms of datasets and tasks. it can be seen that our KuaiSim benchmark is the only one that meets all requirements.
>
> |  Simulators   |   Depth   | Average reward | Total reward |    AUC     |
> | :-----------: | :-------: | :------------: | :----------: | :--------: |
> |     RL4RS     |   14.39   |     0.640      |    9.235     |   0.6929   |
> |    Recogym    |   13.55   |     0.535      |    7.194     |   0.6729   |
> |    RecSim     |   14.05   |     0.588      |    9.347     |   0.6842   |
> | VirtualTaobao |   14.45   |     0.646      |    9.570     |   0.6866   |
> |    KuaiSim    | **14.86** |   **0.679**    |  **10.081**  | **0.7234** |
>
> In Section 6 Table 6 as shown above, we compare our KuaiSim with other simulators quantitatively. On the one hand, we compare the AUC predicted for click signal. KuaiSim has the highest AUC, which shows that KuaiSim is the most accurate simulation of real feedback from users. On the other hand, we utilize the DDPG algorithm to train an agent with different simulators. Among these simulators, KuaiSim outperforms the others by a significant margin across all evaluation metrics. It shows that our KuaiSim has a superior performance in agent training, which indicates enhanced accuracy and reliability of KuaiSim in simulating user behavior. We have refined the text to enhance the clarity of our approach to evaluating simulators in Section 6.
>
> ------
>
> Thank you again for your constructive reviews. Hope that our response can address your concerns. We feel grateful for your appreciation.

---

> > ### Comment · Reviewer_P5ds · 2023-08-18
> > **Thank you for the response**
> >
> > I have no further questions, and I will maintain the score.

---

> > > ### Author Response · Authors · 2023-08-19
> > > **Thank you for your acknowledgment**
> > >
> > > We extend our heartfelt gratitude for acknowledging both our work and our response. We sincerely hope that you could consider endorsing our work during your conversations with other reviewers. Thank you once again.

---

### Official Review · Reviewer_fsvF · 2023-07-21

**Rating:** 6
**Confidence:** 3
**Correctness:** To the best of my knowledge, it is co…

**Strengths:**

1. The proposed simulator is comprehensive to support three important task of recommender systems.
2. Benchmarks of RL-based methods and simulators are conducted on two public datasets.
3. The discussion and comparison regarding existing simulators is clear.

**Additional Feedback:**

Please refer to "Opportunities For Improvement".

**Clarity:**

Yes generally well-written except the over-claiming title. Please refer to "Opportunities For Improvement. 2." for details.

**Documentation:**

No. Documentation of reproducing the proposed benchmarks is not sufficient. There are too many placeholders in the corresponding repository and documents. It can be difficult to follow the code.

**Ethics:**

The checklist for D&B track papers is not associated with the submissions (https://neurips.cc/public/guides/PaperChecklist).

**Opportunities For Improvement:**

1. My biggest concern is that, as a submission to the Datasets & Benchmarks Track, the corresponding GitHub project for this paper has not been well-constructed, making it difficult for audience to understand how to use the proposed simulator, KuaiSim. As of the date of this review, the second section of GitHub's README is completely a placeholder, which raises doubts about whether the simulator, as well as the benchmarks, proposed in the paper is user-friendly, reproducible, and easy-to-follow. (note that the concern has been basically addressed after the rebuttal phase as the authors have updated their GitHub repo)
2. Over-claiming.
    * The proposed simulator is principally designed for RL-based recommender systems. This area isn't typically recognized as a predominant field within recommender system research.
    * The main novelty distinguishing KuaiSim from other existing simulators is its ability to handle additional signals, such as behavioral types and retention.
    * However, it can be difficult for audience to know about the actual contributions from the title "a comprehensive simulator for recommender systems", which appears too general and broad.
3. Presentation issue. "Kuairand" or "KuaiRand"? Please make them consistent in the paper.

**Relation To Prior Work:**

The discussion and comparison regarding prior works is clear.

**Summary And Contributions:**

The paper proposes a more comprehensive simulator designed for RL-based recommender systems, featuring the ability to handle multi-behavior and retention signals. Based on the proposed simulator, the authors set benchmarks for three tasks on two datasets and evaluated different simulators on KuaiRand dataset.

---

> ### Author Response · Authors · 2023-08-18
> **Feedback to Reviewer #2**
>
> We appreciate your overall positive assessment of our contributions and are grateful for your suggestion. We have already updated the new revision based on your suggestions.
>
> ------
>
> **Q1: GitHub project**
>
> **A1:** Your concern is indeed right. Previously, we released an experimental iteration of our project, which has been succeeded by a more user-friendly version (https://github.com/CharlieMat/KRLBenchmark). This updated version explicitly outlines the requisite steps for replicating our results.
>
> To reproduce our results, you can follow the guidelines below:
>
> - Install the necessary packages according to our '0.Setup'.
> - Find the details of data preprocessing in 'code/preprocess/KuaiRandDataset.ipynb'.
> - Run 'run_nultibehavior.sh' to train the simulator.
> - Run 'generate_session_data.sh' to generate session data for agent training.
> - Then you can run the 'train\_(agent name)\_krpure\_(task level).sh' file to train different agents on different tasks with our simulator.
> - We provide visualized training results in 'TrainingObservation.ipynb'.
>
> We have provided detailed parameter settings in all the sh files. For more analysis experiments, you can also check out the project READ ME. And we will be working on further improvements to our project, such as readability guarantees for existing code and the incorporation of novel baselines.
>
> The detaild reproduction guideline is provided in Appendix B in the new revision. We hope our benchmarks could boost future research. The provision of a user-friendly project facilitates result reproduction and the more convenient utilization of KuaiSim. This aspect holds great significance for us. Once again, we appreciate your insightful suggestions.
>
> ------
>
> **Q2: Over-claiming**
>
> **A2:** Thank you for your constructive suggestions.
>
> On the one hand, non-RL recommender systems generally don't use simulators. Directly deploying RL models in online environments turns out to be challenging and resource-intensive, since an untrained or premature recommendation model can adversely impact the user experience and lead to undesirable real-time data, subsequently affecting the model's training performance. In order to mitigate these challenges, the research of user simulators has emerged recently to serve as a pre-online verification method for RL models.
>
> On the other hand, our KuaiSim boasts the distinct advantage of encompassing three levels of recommendation tasks, which covers a broad spectrum of recommendation problems. Additionally, KuaiSim also models broader and more heterogeneous user feedback. Compared to other simulators, as shown in Table 1 in Section 2.4, RecoGym and RecSim are not for real dataset, and many existing competitive simulators are limited to single-task support, lacking coverage across all tasks. In contrast, KuaiSim stands out by fulfilling comprehensive demands, supporting a wide range of tasks. Our title wants to highlight this contribution.
>
> Of course, there are some recommendation problems, user feedback, and domain uncovered by our simulator. However, KuaiSim holds the potential to overcome these limitations. For uncovered recommendation problems, such as explainable recommender systems, diversity recommendations, and fairness recommendations, KuaiSim can broaden its scope by incorporating the interpretability, diversity, fairness, and other indicators of the recommendation results into the simulation process. Uncovered user feedback can be easily incorporated through supervised training. For uncovered domain, the iterative refinement process we've undertaken for constructing and evaluating the simulator greatly facilitates such domain expansions. We claim these limitations and potential solutions in Section 6 of our new revision.
>
> Overall, our KuaiSim effectively addresses a wide array of critical recommendation challenges while incorporating extensive user feedback. Its adaptability to diverse scenarios, as demonstrated in our research, underscores its status as a comprehensive simulator. We hope our response could address your concerns. If there still remains any consideration, please kindly let us know. We are very happy to make a further revision in light of your great suggestions.
>
> ------
>
> **Q3: Presentation Typo `Kuairand’**
>
> **A3:** We conduct experiments on KuaiRand dataset, which is misspelled as Kuairand in some paragraphs. We apologize for any confusion caused by this typo. We have modified this typo and carefully checked our paper and reorganized the presentation.
>
> ------
>
> Thank you again for your constructive reviews. Hope that our response can address your concerns. We will feel grateful if you could boost our paper.

---

> > ### Comment · Reviewer_fsvF · 2023-08-19
> > **Thank you for the constructive response**
> >
> > Thank you for the detailed response and efforts towards updating the GitHub repo. Most of my concerns have been addressed. I'll raise my initial rating.

---

> > > ### Author Response · Authors · 2023-08-19
> > > **Thank you for your acknowledgment**
> > >
> > > We greatly appreciate your acknowledgment of our efforts and response. Additionally, we are truly grateful for raising our rating. Thank you immensely!

---

### Official Review · Reviewer_uGYD · 2023-07-21
**A comprehensive and powerful simulator for reinforcement learning-based recommender systems**

**Rating:** 6
**Confidence:** 3

**Strengths:**

(1) The paper addresses an important gap in existing simulators, notably in terms of multi-behavior and cross-session user responses, bringing it closer to real-world environments.
(2) The use of log data to pretrain user response models enhances the reliability of KuaiSim in replicating real-world scenarios.
(3) The authors also benchmarked their simulator against several popular recommendation algorithms, offering a comprehensive performance evaluation.


**Additional Feedback:**

The authors might want to include a discussion on the potential real-world applications of KuaiSim and how it might help organizations improve their recommendation systems.

**Clarity:**

The paper is well written and clearly structured. The use of technical jargon is appropriate and does not obscure the presented ideas. The authors have also provided ample references for further understanding.

**Correctness:**

The claims in the paper appear to be well-grounded, supported by an extensive evaluation on two datasets. The methodology used for the construction of the simulator is sound, and the benchmarks used for evaluation are relevant.

**Documentation:**

The paper provides sufficient detail on data collection, organization, and responsible use. The code is available online, enhancing reproducibility.

**Ethics:**

The authors have not discussed potential ethical implications in detail, especially in terms of data privacy. As the simulator is trained on log data, discussion around the anonymization of user data and ensuring privacy should be included.

**Limitations:**

The authors did not report any limitation and potential negative societal impact of their proposed KuaiSim. The societal implications of the technology and how it interacts with user data privacy, are not adequately addressed in the paper.

**Opportunities For Improvement:**

1) Lack of Discussion on Limitations: One of the main areas for improvement is the need for an explicit acknowledgment and discussion of the limitations of the research. Every research work has its limitations, and it is crucial to recognize and disclose them. This adds credibility to the study and provides a pathway for future research.

2) Ethical Considerations and Societal Impact: Considering the paper used real user logs to train the simulator, it's necessary to include a detailed discussion about the ethical considerations and potential negative societal impact. This can involve aspects such as privacy concerns regarding user data, potential misuse of trained models, and broader societal implications.

3) Absence of Error Bars in the Reported Data: Another critical point that needs addressing is the omission of error bars in the experimental data presented in the tables. Error bars are essential in understanding the variability or uncertainty in the data. Not including them could affect the reproducibility and reliability of the results. It would be beneficial for the authors to provide these metrics in any future research or revisions.

**Relation To Prior Work:**

The authors have clearly delineated how KuaiSim differs from existing simulators. They have adequately compared and contrasted their work with prior research and simulators, highlighting the strengths and weaknesses of each.

**Summary And Contributions:**

This paper presents KuaiSim, a comprehensive simulator for reinforcement learning-based recommender systems. It addresses three levels of recommendation tasks: list-wise, sequential, and retention optimization. It provides a robust alternative to existing simulators, offering multiple behavior and cross-session responses, which are essential to real-world environments. The paper offers evaluation protocols, benchmarks, and comparisons with other competitive simulators. In addition, it showcases the data migration capabilities of KuaiSim on the public dataset ML-1m.

---

> ### Author Response · Authors · 2023-08-18
> **Feedback to Reviewer #1**
>
> We appreciate your overall positive assessment of our contributions and are grateful for your suggestion. We have already updated the new revision based on your suggestions.
>
> ------
>
> **Q1: Lack of discussion on limitations.**
>
> **A1:** Thank you for your valuable suggestions. Our work exhibits deficiencies in certain areas. We conclude them as follows, and give potential direction for solutions:
>
> -  **Uncovered recommendation problems** Compared to previous works, our simulator provides a user response environment at three distinct task levels, which contains most recommendation problems. But it's important to acknowledge the existence of numerous other challenges that warrant attention, such as explainable recommender systems, diversity recommendations, and fairness recommendations.
> -  **Expanding to diverse domains** While both the KuaiRand and ML-1m datasets center around video recommendations, it's worth noting that the KuaiSim framework can be extended to encompass a variety of domains such as music, news, and e-commerce. The iterative refinement process we've undertaken for constructing and evaluating the simulator greatly facilitates such expansions.
>
> A more detailed description is provided in Section 6 in the new revision.
>
> ------
>
> **Q2: Ethical considerations and potential societal impact.**
>
> **A2:** We categorize our ethical considerations and potential societal impact into three key aspects.
>
> **Safeguarding user privacy:** While our approach involves training the simulator using real user logs, it's important to note that the datasets we employed have been stripped of any privacy-sensitive information. The user characteristics utilized in our study are comprehensively presented in Appendix A.1 Table 2, shown as follows:
>
> | User feature          | Feature type | Explanation                                              |
> | --------------------- | ------------ | :------------------------------------------------------- |
> | video id              | int64        | The unique identifier for the user.                      |
> | user active degree    | str          | The level of user activity.                              |
> | is live streamer      | int64        | Indicates whether the user is a live streamer.           |
> | is video author       | int64        | Indicates if the user has uploaded any videos.           |
> | follow user num range | str          | The range of the number of users that this user follows. |
> | fans user num range   | str          | The range of the number of fans of this user.            |
> | friend user num range | str          | The range of the number of friends that this user has.   |
> | register days range   | str          | The range of the number of days since user registration. |
> | one-hot features      | int64        | Encrypted features of the user.                          |
>
> These characteristics reflect the user's interactions with the application, and each user is denoted by an anonymous uid. Thus, it's essential to emphasize that we do not leverage any private user information, effectively mitigating concerns related to privacy breaches.
>
> **Unbiased KuaiRand dataset:** Because online A/B test usually consumes much time and money, which makes it impractical for academic researchers to conduct the evaluation online. However, offline evaluation will cause bias because of massive missing data, i.e., the user-item pairs which have not occurred in the test set. One fundamental approach to address this issue in offline evaluation is to collect unbiased data, i.e., to elicit user preferences on the randomly exposed items. This unbiased dataset empowers us to conduct unbiased offline evaluations without compromising user privacy. The detailed data collection process can be found in Appendix A.1.
>
> **Promoting field advancement:** We provide various competitive algorithms for three task levels as benchmarks, coupled with a meticulously enhanced process for constructing and evaluating simulators. This framework not only facilitates the advancement of the field but also presents a promising avenue for research by extending these simulators to encompass a broader spectrum of tasks and fields.
>
> A more detailed description is provided in Section 6 in the new revision.
>
> ------
>
> **Q3: Absence of Error Bars in the Reported Data.**
>
> **A3:** All our results are averaged over five distinct sets of results to ensure the reliability of the results. And we add error bars for all reported data, where $\pm$ represents the standard deviation of five results. We make this clear in Section 5 in the new revision. For the comparison between KuaiSim and other simulators in Table 6, we also employ a t-test. Providing reproducible and reliable benchmarks is our goal, thanks for your constructive suggestions.
>
> ------
>
> Thank you again for your constructive reviews. Hope that our response can address your concerns. We will feel grateful if you could boost our paper.

---

> > ### Comment · Reviewer_uGYD · 2023-08-27
> >
> > Thanks for your prompt response. Most of my concerns have been addressed and I will maintain the score.

---

### Author Response · Authors · 2023-08-18
**General response**

We thank all reviewers for their insightful comments and suggestions. We are particularly encouraged by the reviewers’ feedback. We have made a heavy revision to our paper according to the reviewer's constructive suggestions. Below we summarize some key modifications in this revision:

-  A discussion on the limitation in Section 6 (Reviewer #1,  Reviewer #2):
   -  Uncovered recommendation problems

   -  Expanding to diverse domains

-  A discussion on ethical considerations and potential societal impact in Section 6  (Reviewer #1):
   - Safeguarding user privacy
   - Unbiased KuaiRand dataset
   - Promoting field advancement

-  More detailed experiments settings
   - Error Bars in Section 5 (Reviewer #1)
   - Dataset splits in Section 3.3  (Reviewer #3)

-  GitHub project (Reviewer #2)
   - We have released a user-friendly version, which explicitly outlines the requisite steps for replicating our results
   - We provide detailed reproduction guidelines in Appendix B

-  Sensitivity analysis (Reviewer #3)
   - We conduct parameter sensitivity analysis for two vital parameters max time step and slate size in Section 6

-  Checklist (Reviewer #3)
   - We add the checklist after the references section and check our paper against the checklist

Moreover, we have carefully checked our paper typos and reorganized the presentation. If there still remains any consideration, Please kindly let us know. We are very happy to make a further revision in light of your great suggestions. We will address comments by each of the reviewers individually.

---

### Decision · Program_Chairs · 2023-09-22

**Decision:**

Accept (Poster)

**Comment:**

The paper proposes a simulator for RL-based recommender systems, using the datasets from Kuaishou. Evaluation protocols, benchmarks, and comparisons with other simulators are offered. All reviewers agree that the proposed simulator is comprehensive and simulates the real-world environments well. The discussion and comparison regarding existing simulators is clear. Some reviewers also express their concerns on ethical considerations and societal impacts as the simulator used real user logs to train the simulator. The discussion addressed some of the concerns received during the initial review round.